# Effect of Sealing Treatment on Corrosion Resistance of Arc-Sprayed Zn and Zn85-Al15 Coatings

**Bo Li** [1], **Zhuoyi Liu** [2], **Jinhang He** [1], **Jie Bai** [1], **Haibo Jiang** [3], **Ye Tian** [4], **Zhiqing Zhang** [4] and **Shifeng Liu** [4,*]

1 Electric Power Research Institute, Guizhou Power Grid Co., Ltd., Guiyang 550000, China
2 Guizhou Electric Power Grid Dispatching and Control Center, Guiyang 550002, China
3 Duyun Power Supply Bureau, Guizhou Power Grid Co., Ltd., Duyun 561000, China
4 College of Materials Science and Engineering, Chongqing University, Chongqing 400044, China
* Correspondence: liusf06@cqu.edu.cn

**Abstract:** This study investigated the corrosion morphology of arc-sprayed Zn and Zn85-Al15 coatings with and without sealing treatment under simulated defect conditions. The hole sealing treatment was carried out by Conventional Impregnation Sealing (CIS). The performance of two coatings was assessed by employing morphological analysis, chemical composition, and electrochemical studies. The results showed that the corrosion performance of two coatings with sealing treatments was better than that of the un-sealing coating. Through the double protection of coating and sealing treatment, the defect-free material has excellent anti-corrosion performance in the salt spray experiment. In the simulated defect environment, the Zn-Al coating has better corrosion resistance, and the corrosion products are denser and more stable near the defects, which reduces the overall corrosion rate of the coating. The electrochemical experiment results demonstrated that the Zn-Al coating exhibited a three times lower corrosion rate compared to the Zn coating in the immersion experiment, and the corrosion rate did not change with the thickness of the coating. The average adhesion values of the two different methods (chilled iron and alumina) were similar (~8 MPa), but after the salt spray test, the adhesion strength increased. The corrosion mechanisms of Zn coating and Zn-Al coating are analyzed and compared. In general, it was indicated that the Zn85/Al15 coating prepared by sealing treatment has better corrosion resistance than the matrix steel. This study can provide some reference for corrosion behavior in defect environments.

**Keywords:** corrosion resistance; Zn and Zn85-Al15 coatings; electrochemical corrosion; salt spray

## 1. Introduction

With the development of industry, the atmospheric content of acidic species is increasing rapidly. Metallic facilities are at high risk of failure due to acid rain corrosion, especially for steel materials, so corrosion protection has received more and more attention [1–4]. For a long time, people have been using coating, electroplating, hot dip plating, and other methods to prepare coating for corrosion protection, among which thermal spraying technology has been considered the most economical and effective protection method [5]. Arc spraying is a process in which the arc is used as the heat source and the metal wire is melted and atomized by air flow so that the molten particles are sprayed to the material surface at high speed to form the coating [6]. The arc spraying of zinc and zinc-aluminum alloy coatings has a good application effect in the anti-corrosion engineering of steel matrix components, making these coatings a popular surface treatment method to protect steel structures in industrial and marine environments [7]. By spraying zinc on the substrate steel, the zinc coating has electrochemical activity, which can effectively protect the substrate steel by sacrificing the anodic zinc coating. Previous researchers have performed a lot of research on the corrosion resistance of Zn and Zn/Al alloy coatings. For example, Liu et al. [8] simulated the corrosion behavior of Zn coatings in atmospheric exposure experiments

in typical tropical marine areas and found that they have a certain degree of corrosion protection. Zhou et al. [9] designed and prepared pure Zn coating A by thermal spraying and pore-sealing coating B by resin spraying on the surface of Q235 steel substrates and carried out corrosion experiments on the two coatings by tap water immersion. The experimental results show that the thickness of pure Zn coating A by thermal spraying is 281 μm, and the porosity is about 11.8%. The thickness of composite coating B is 287 μm, and the porosity is about 7.2%. However, the thermally sprayed Zn coating has relatively high porosity, is very sensitive to Cl$^-$, and is prone to pitting. The corrosion of Zn-based coatings is mainly uniform corrosion, which can protect the substrate well even if it is very thin [10–12]. Wang et al. [13] used arc spraying technology to spray Zn coating on the surface of Q235 steel and to simulate an acid rain corrosion environment. The results show that the surface corrosion of the Zn coating will be aggravated by the decrease in pH value of the simulated acid rain solution. When the pH value of the simulated acid rain solution is between 5 and 2.8, the corrosion of the Zn coating is more serious. Furthermore, Zn-Al coatings have been introduced to protect the steel matrix. For example, Zhao et al. [14] prepared the Zn-Al alloy coating on the surface of Q235 substrate by mechanical method, and the alloy coating had a better inhibition effect on charge transfer. The corrosion product was dense, which can enhance the physical shielding function. Chen et al. [15] prepared the electrogalvanizing coating and the Zn-Al coating on the cold-rolled plate substrate and conducted a 42-day salt spray test. The results showed that the corrosion resistance of the zinc-aluminum coating was stronger than that of the electrogalvanizing coating, and the electrogalvanizing coating completely failed after 28 days of cycle salt spray testing, while the Zn-Al coating did not fail after 42 days of testing. Zhang et al. [16] prepared three kinds of Zn-Al coatings with high aluminum content using thermal spraying technology. The results show that zinc is first dissolved in the corrosion process of Zn-containing coatings, and the corrosion products attached to the coating surface make the coatings show a weak self-sealing effect. Zn-Al60 coatings have both the cathodic protection effect of zinc and the passivation shielding effect of aluminum, showing the best corrosion resistance among the three coatings. Sui et al. [17] studied the anti-corrosive properties of two kinds of Zn-powder content in hot-sprayed Zn-Al coatings in seawater at different temperatures from 0 °C to 25 °C. The results show that with the decrease in temperature, the corrosion potential of the coating is positive, the corrosion current density is decreased, the coating resistance (Rc) is increased, and the corrosion resistance of the coating is increased. In general, an Al-2%Zn (mass fraction) coating has better corrosion resistance and stability than an Al-85%Zn coating in low-temperature seawater. Despite the fact that a substantial amount of work has been performed in the field of the corrosion behavior of thermally sprayed sacrificial coatings, for most coatings, corrosive media such as air, water, and ions can directly corrode the substrate through the hole and cause the coating to fall off and fail. For example, when the alloy coating is contaminated with salt, the oxidation rate is faster at a certain temperature. As a result, the protective layer breaks down and salts reach the metal surface, causing degradation and failure [18]. Although experiments on zinc and zinc-aluminum coatings have been carried out, no further studies have been conducted on the porosity of coatings. The porosity of the coating directly determines its corrosion resistance. There are a lot of pores between the coating material and the matrix metal, forming a loop when used, which reduces the protective effect. However, the pore-sealing agent can effectively close most of the through holes in the coating, block the corrosion channel, and significantly improve the corrosion resistance of the coating. Therefore, the performance of the coating after sealing is also a new research hotpot. Through the sealing treatment, some pores, micro-pores, micro-cracks, and other defects existing in the coating can be filled and closed, forming a thin layer above the coating and thus improving the corrosion resistance and other properties of the coating [19]. Among them, the sealing treatment technology is mainly on the surface of the coating, through brush coating, impregnation, and other ways to achieve the purpose of sealing pores [20]. In addition, the bonding strength can be improved by arc spraying surface treatment. Jang et al. [21] evaluated the

metal spraying efficiency, bonding properties, and bonding strength under different surface treatment conditions. However, the application of coatings goes far beyond that [22,23]. XU analyzed and compared the effect of defects in arc spraying zinc-aluminum coating on coating performance and improved the corrosion prevention effect after sealing the coating [24]. Moshtaghi et al. [25] studied the microstructure of electrodeposited ZnNi coatings and analyzed the hydrogen diffusion behavior caused by the existence of defects (pores and microcracks). As can be seen from the above literature, most researchers have analyzed the corrosion conditions in the corrosive environment and the effect of sealing treatment on the corrosion resistance of Zn-Al alloy coatings. Few researchers analyze the defects combined with the corrosive environment. It is not comprehensive to consider only the corrosive medium without considering the defect problem. There is a certain difference in the actual application situation. In practice, corrosive media and defects (gaps and holes) are mutually present. The analysis of the corrosion mechanisms of single-layer and composite coatings is less thorough. Therefore, it is of great value to systematically study the hole sealing performance of arc spray Zn and Zn/Al coatings for long-term corrosion protection, simulate the defect environment, and analyze the corrosion mechanism of coatings.

In this paper, the sealing treatment of Zn and Zn/Al coatings with different thicknesses was carried out, and at the same time, the defect environment of the coating was simulated. The salt spray test and full immersion test were used to compare the coating properties before and after the treatment with the sealing agent. The properties of Zn and Zn/Al coatings after treatment with sealing agents were comprehensively compared. After the sealing treatment, the corrosion resistance of the Zn or Zn/Al alloy coating is greatly improved. The experiment simulated an acid rain environment by configuring corrosion solutions with different concentrations and demonstrated the corrosion performance and mechanism of the two coatings by simulating defects. Through the experiment, the spraying protection of matrix steel is realized and its long-life application is realized, which has a good reference value for the protection of matrix steel.

## 2. Materials and Methods

### 2.1. Sample Preparation

The matrix material was S355 steel, and the chemical composition is shown in Table 1. The coating materials were pure Zn and pure Al. The Zn wire and Al wire were placed in different spray guns, respectively. For the preparation of single Zn coatings, only the Zn wire gun is used. For the preparation of the Zn-Al coating, two spray guns are used.

**Table 1.** Chemical composition of S355 (wt%).

| Element | C | Si | Mn | P | Ni | Nb | Ti | Fe |
|---|---|---|---|---|---|---|---|---|
| (wt%) | 0.11 | 0.19 | 1.44 | 0.011 | 0.19 | 0.38 | 0.15 | Bal. |

All coatings were prepared at TWI on S355 steel using a mechanized Metallisation Arc 140 twin wire arc spraying (TWAS) system, as shown in Figure 1e. Before spraying, the sample was cleaned with acetone, and then an ultrasonic treatment was performed. Carbon steel (S355) substrates (two different sizes: 150 mm × 100 mm × 5 mm for salt spray, 50 mm × 50 mm × 5 mm for immersion tests) were prepared. The spray sample is shown in Figure 1a–d. Two different grit types were employed for the surface preparation (chilled iron and alumina) to enhance the bonding strength between the coating and the matrix. The optimized parameters are shown in Table 2. The thickness of the coating is 150 mm and 300 mm, respectively. The hole sealing treatment was carried out by Conventional Impregnation Sealing (CIS). One set of specimens was sealed/overcoated with a basecoat of Sigmafast 278 (~0.25 mm dry film thickness (DFT)) and a top coat of Sigmadur 520 (~0.075 mm DFT). The corresponding experimental parameters are shown in Figure 2. In order to simulate a more realistic defect environment, the specimens were also scribed (50 mm × 2 mm) before environmental testing, as shown in Figure 1. The scribe was

produced to cut through the coating all the way to the substrate. The schematic diagram of the experimental process is shown in Figure 2. In order to ensure the reliability of the data, three sets of parallel tests were used for analysis.

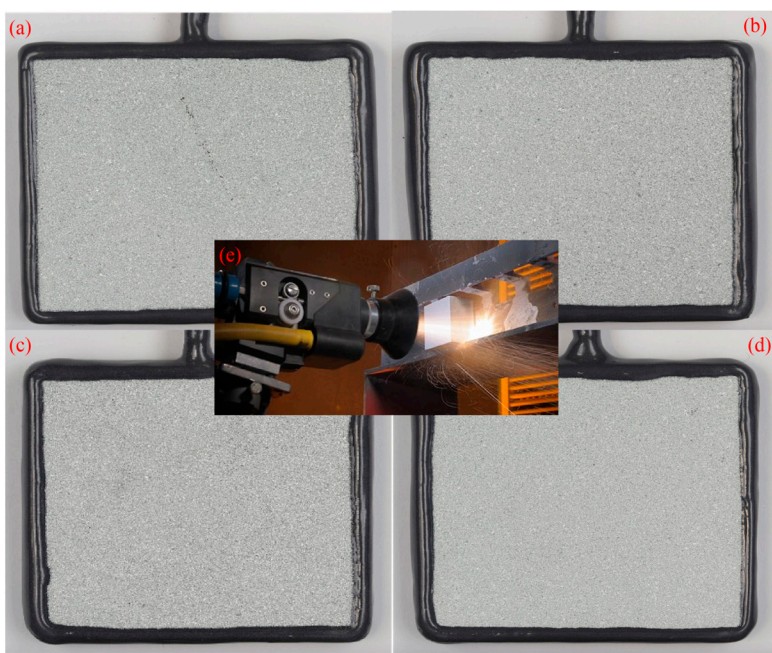

**Figure 1.** Schematic diagram of arc spraying and samples (**a**) 0.15 mmZn, (**b**) 0.30 mmZn, (**c**) 0.15 mmZn/Al, (**d**) 0.30 mmZn/Al, (**e**) Spray diagram.

**Table 2.** Technological parameters used in arc spraying.

| Coatings | Wire Diameter (mm) | Speed (mm/s) | Voltage (V) | Coating Thickness (μm) | Paint Layer Thickness (mm) |
|---|---|---|---|---|---|
| Zn | 2.3 | 500 | 28 | 150/300 | 0/0.325 |
| $Zn_{85}$-$Al_{15}$ | 2.3 | 500 | 28 | 150/300 | 0/0.325 |

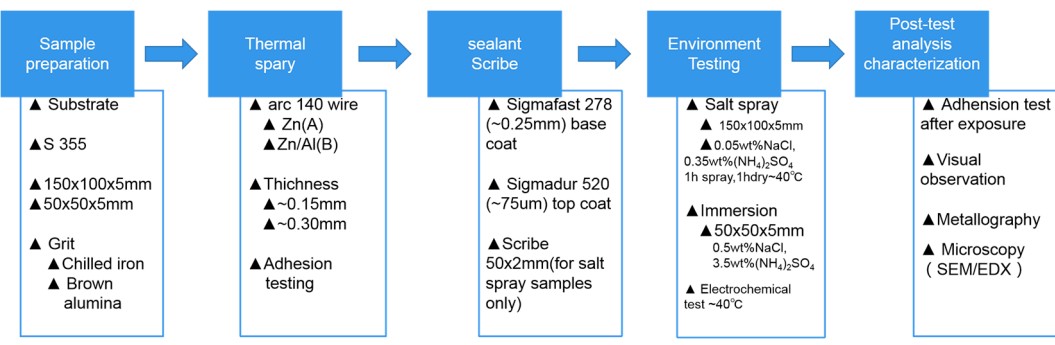

**Figure 2.** The schematic diagram of the experimental process.

### 2.2. Salt Spray Test

Salt spray specimens (150 mm × 100 mm × 5 mm) were taken, and uncoated surfaces (back, sides, and edges) were covered with a polymeric resin to define the test surface. The salt spray test was conducted using a solution containing 0.05 wt% NaCl and 0.35 wt% $(NH_4)_2SO_4$. The samples were placed in a salt spray chamber and exposed to alternate wet cycles consisting of 1 h of spray and 1 h without spray at ~40 °C. The test was carried out for 3 months. Figure 3e shows the salt spray test equipment.

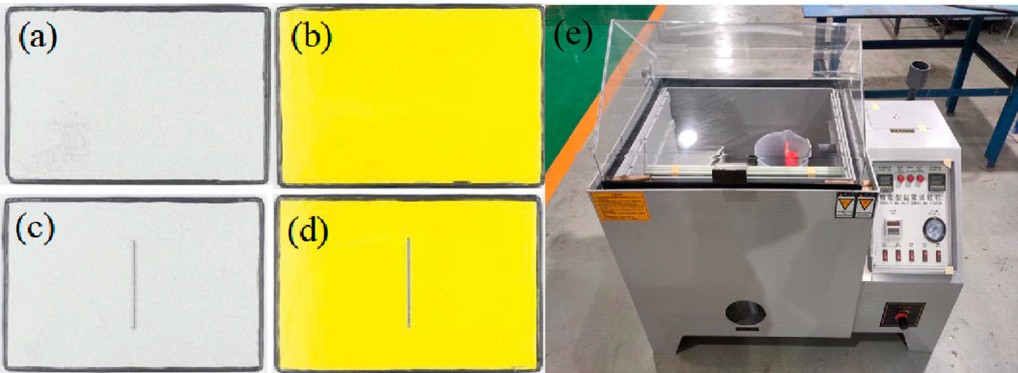

**Figure 3.** (**a**) Zn coating without scribe; (**b**) Zn coating with top paint layer; (**c**) Zn coating with scribe; (**d**) Zn coating with top paint layer and scribe; (**e**) The salt spray chamber.

*2.3. Immersion Test*

　　An electrochemical corrosion test was conducted according to GB/T 24196-2016. The potentiodynamic polarization curve test equipment was a CS310 electrochemical workstation. The classical three-electrode system was adopted; the reference electrode was a saturated calomel electrode, and the Pt electrode was used as the auxiliary electrode. Similar to the salt spray specimens, the uncoated surfaces of the immersion test specimens (50 mm × 50 mm × 5 mm) were also covered with a polymeric resin. Electrical connections were made on the specimens before insulating them with the resin, the exposed area is 50 mm × 50 mm. The electrolyte contained 0.5 wt% NaCl and 3.5 wt% $(NH_4)_2SO_4$. The test solution was maintained at ~40 °C, and the specimens were tested under continuous immersion conditions for 3 months. Thermal spray coatings subjected to immersion testing with electrochemical monitoring were evaluated in the unsealed condition only. To allow electrochemical measurements, a three-electrode setup was used. The specimen to be tested was made the working electrode, a Pt (Ti) wire was used as the counter electrode, and Ag/AgCl (Sat. KCl) was used as the reference electrode. The open circuit potential (OCP) of the specimens was measured against the reference electrode using an ACM potentiostat. The linear polarization resistance (LPR) technique was used to measure the polarization resistance as a function of time. Potentiodynamic polarization tests were performed after 1500 s of open-circuit monitoring. An overpotential of −0.2 to 0.2 V (v.s. OCP) was applied with a scanning rate of 0.3 mV/s. To ensure reproducibility, all tests were performed at least three times. The slope of the polarization curve vs. current density at zero polarization gave polarization resistance ($R_p$). A potentiodynamic polarization test was carried out after the completion of the exposure to give Tafel parameters. Tafel analysis was carried out to find the anodic ($b_a$) and cathodic ($b_c$) Tafel slopes from the potentiodynamic polarization curves. The incorporation of the Tafel slopes and $R_p$ in the equation (i) below gave the corrosion current ($I_{corr}$) [26].

$$I_{corr} = \frac{b_a \times b_c}{2.303 \times (b_a + b_c)} \times \frac{1}{R_p} \tag{1}$$

　　The calculated corrosion current ($I_{corr}$) was converted to a corrosion rate (CR) by using the equation (ii) below [27,28]:

$$CR = \frac{I_{corr}}{\rho} \times EW \times K_1 \tag{2}$$

where 'ρ' is the density of the alloy (Zn = 7.13 g/cm³, Zn-Al = 6.47 g/cm³) and EW is the equivalent weight (Zn = 32.68, Zn-Al = 23.43). CR is given in mm/y when $I_{corr}$ is in μA/cm² and $K_1 = 3.27 \times 10^{-3}$ mm g/μA cm y. These values were taken from ASTM G1 and ASTM G102.

### *2.4. Post-Test Analysis and Characterization*

Following testing, coating analysis and characterization took various forms. The macroscopic morphology of corrosion in the two coatings was analyzed and recorded using photography technology. The coating defects and surface morphology of the as-sprayed coatings and coatings after salt spray were examined using a field emission scanning electron microscope (JEOL JSM-7800F, Tokyo, Japan). This instrument was equipped with an energy-dispersive X-ray spectroscopy (EDS) tool for elemental analysis. The errors in SEM and EDS are small, which has little influence on the final analysis result. Coating adhesion (pre-test and post-test) performed in accordance with ASTM D4541 is reported in terms of tensile bond strength.

## 3. Results and Discussions

### *3.1. Macroscopic Morphology Analysis*

As can be seen from the macroscopic morphology diagram, after 3 months of salt spray testing, a large number of corrosion rusts appear in the matrix steel without any treatment, as shown in Figure 4c. The corrosion products completely cover the steel matrix and are distributed uniformly on the surface of the steel matrix. Previous studies have shown that at the initial stage of salt spray, the matrix steel is dissolved first, and Fe mainly exists in the form of $Fe^{2+}$, $Fe(OH)_2$, and $Fe(OH)^+$. With the extension of time, $Fe(OH)_2$ and $Fe(OH)^+$ continue to interact with the corrosive medium to form iron compounds [29,30]. For the Zn coating with no defects and a painted surface, the coating samples showed different degrees of corrosion after three months of salt spray testing, and the white corrosion products were distributed on the surface of the coating. For Zn coatings with defects and no paint surface, after three months of salt spray testing, a large number of white corrosion products appeared near the defects, while more dense pitting products appeared far away from the defect region. For the Zn coating with defects and a painted surface, after three months of salt spray testing, a large number of white corrosion products only appear near the defects in the coating sample, and no obvious corrosion phenomenon occurs in the area far from the defects. For the defect-free and paint samples of Zn coating, after three months of salt spray testing, no corrosion products appeared in the coating samples, indicating that the paint has good protection against the defect-free samples. Similar results were found for Zn-Al coatings. The morphology of the sample with and without scribe has no great influence, while without paint, the scribe regions in the paint are corroded to different degrees. A specific discussion was carried out regarding microstructures. In general, defect-free material has a good protection mechanism for the base material under the dual protection mechanism of coating and sealing treatment. It can be seen from the figure that there is no obvious bulge phenomenon in the coating with or without the sealing treatment. Xiao et al. [31] made a similar report on zinc coating before and after sealing treatment.

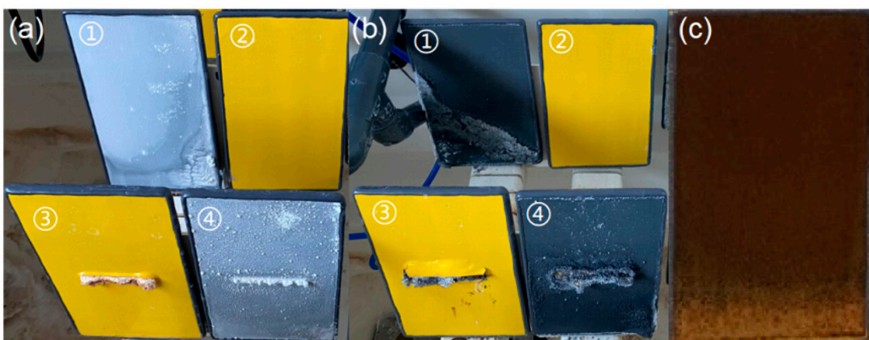

**Figure 4.** Specimens in the salt spray chamber after 3 months of testing. (**a**) Zn coating ①: the coating without paint and defects; ②: the coating with paint and without defect; ③: the coating with paint and defect; ④: the coating with defect and without paint. (**b**) Zn/Al coating ①: the coating without paint and defect; ②: the coating with paint and without defect; ③: the coating with paint and defect; ④: the coating with defect and without paint. (**c**) The matrix steel after corrosion.

*3.2. Microscopic Morphology Analysis*

3.2.1. Microstructure of the Zn Coating

In order to simulate the corrosion environment, the microstructure analysis of the defective sample was carried out. The micrographs of cross-sections reveal the corrosion products formed after exposure of Zn coatings (without paints) to the corrosive environment. The microstructure of the coating is relatively dense, but there are some through pores in the coating. The corrosion products formed an envelope over the scribe region, as shown in the red area (Figure 5a). These corrosion products contained Zn, O, S, and Na as the main constituents, as seen in region 1 with the EDS pattern (Figure 5b). Similar corrosion products were also seen in region 2, away from the scribe (Figure 5c). Some Au peaks are also evident, as a thin layer of Au was sputtered onto the specimens prior to SEM examination. In addition, there is evidence of corrosion damage in the Zn coating. The corrosion products are mainly distributed in the longitudinal direction and less in the transverse position, which is mainly the result observed in the cross section. In addition to the copious amount of Zn-corrosion products, the high magnification image shows that the coating has developed micropits (Figure 5d). This type of micropit seems to be present in the coating. The existence of holes will lead to the formation of corrosion through holes; the corrosive solution will enter the matrix through holes and accelerate the corrosion process.

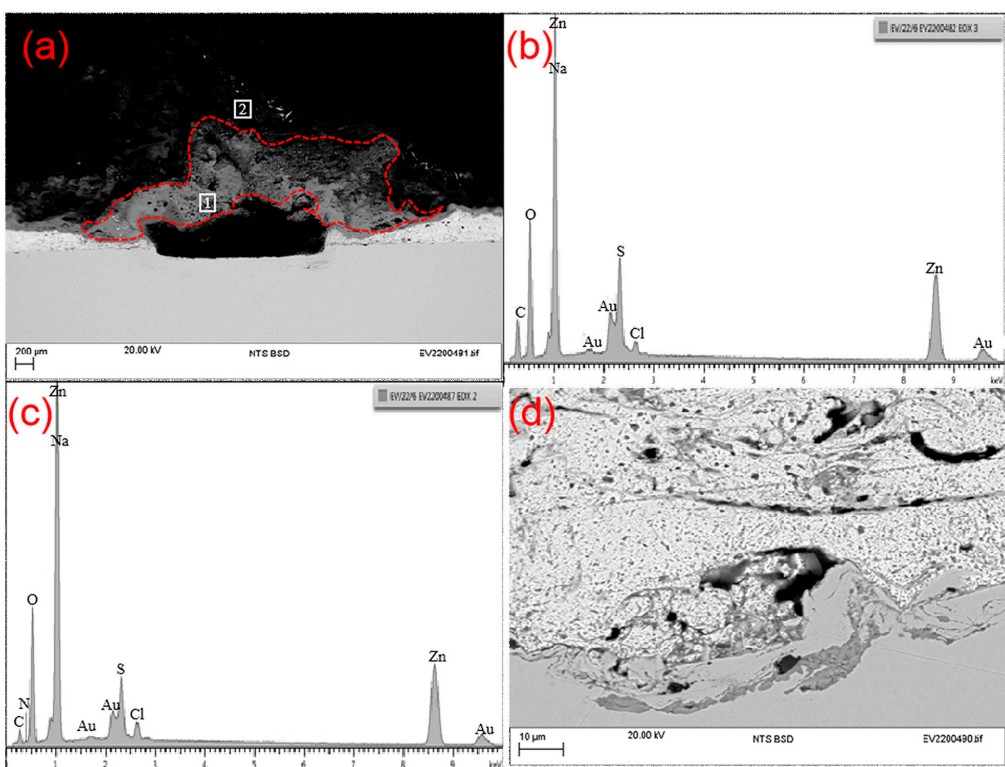

**Figure 5.** SEM micrograph of a Zn-coated specimen showing microstructural features after 3 months of testing. (**a**) The microscopic region of the scribe region; (**b**) energy spectrum analysis of region 1; (**c**) energy spectrum analysis of region 2; (**d**) the high magnification image of (**a**).

Figure 6 shows the Zn coating on the paint surface. It can be seen from Figure 6c that the coating, substrate, and paint surface have a good binding effect. As a contrast, the performance of the Zn coating overcoated with paint depends on the location. As can be seen from Figure 6, after the sealing treatment, the O strength of the longitudinal section area increases significantly compared with the surface area, while the strength of the S and Cl elements decreases, indicating that the corrosive medium does not penetrate into the matrix in this environment. Liu et al. studied the morphology and properties of the Fe matrix after sealing. The experimental results show that the porosity of the Fe

matrix decreases after sealing, which can enhance its anti-corrosion performance [32]. The region close to the scribe shows voluminous deposits of corrosion products, as shown in Figure 6a. The scribed region is completely covered with deposits. Figure 6b shows the EDS pattern of corrosion products near the scribe defect; the deposits contain Zn, O, Na, and S, and the experimental results are similar to those of Cui et al. [33]. The region close to the substrate surface shows corrosion products containing Fe, as shown in Figure 6d. The regions away from the scribe show very little effect of exposure, particularly on the top surface. The paint layer looks largely intact, as shown in Figure 6c. The regions closer to the substrate, however, show features indicative of pitting damage. Compared to the sample without paint, it has a better protection effect away from the scribed region. Under the dual protection of Zn coating and sealing treatment, it has a good anti-corrosion effect far away from the scribed region.

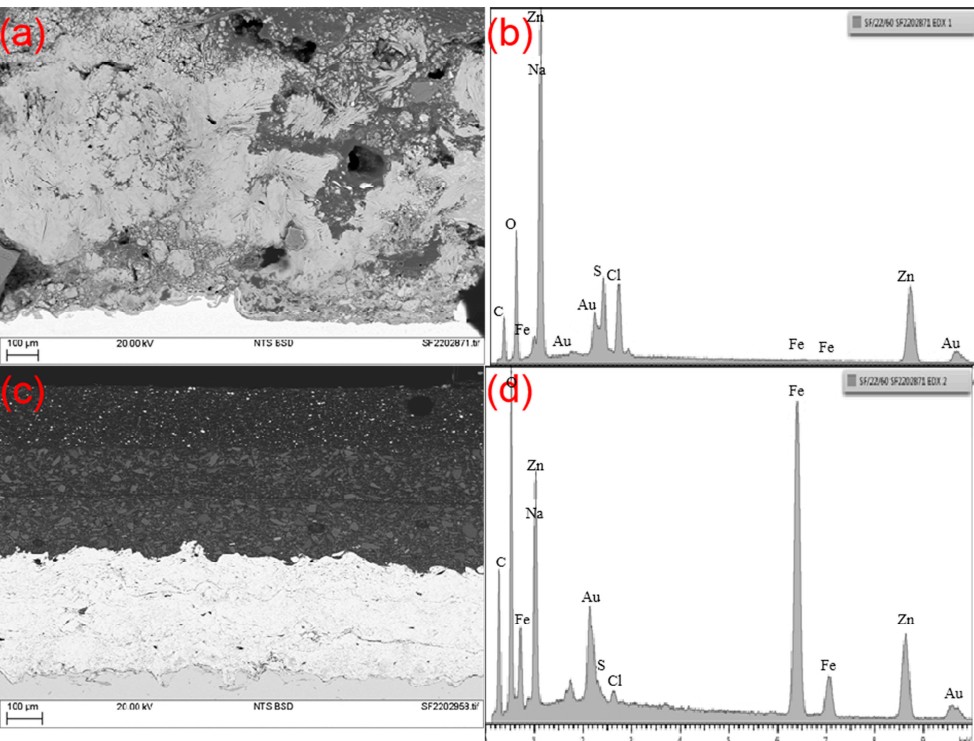

**Figure 6.** SEM micrograph of a Zn-coated specimen with paint showing microstructural features after 3 months of testing. (**a**) SEM of the top; (**b**) the EDS pattern near the scribe defect; (**c**) SEM of the cross-section; (**d**) the EDS pattern close to the substrate.

### 3.2.2. Microstructure of the Zn-Al Coating

Figure 7 shows a cross section of an unpainted Zn-Al coating exposed to a corrosive fluid for 3 months. It can be seen that extensive damage has developed in the scribed region. The coatings around the scribed region have been almost consumed, with little evidence of coating left, as shown in Figure 7a. A large number of corrosion products are produced in this area. As the coating of the scribed region is destroyed, the base material is exposed, as shown in Figure 7b; almost all around the scribed region is corroded, and relatively slight corrosion is present away from the scribed region. Figure 7c shows the EDS analysis of the corroded area. It can be seen from the figure that this region includes Fe, Zn, O, C, and some of the constituents of the corrosive media, such as S, Na, and Cl. Figure 7d is the SEM of an area far away from the scribe region, and it can be seen from the figure that the areas further away from the scribe also show extensive corrosion damage. The top layer of the coating has significantly corroded. Wang et al. reported similar results [34]. Simultaneously, it would appear that the corrosion products are distributed in the coating. The EDS analysis shows the presence of Zn, Al, O, and C as the major constituents of this

corroded layer, as shown in Figure 7e. According to Figure 7f, with higher magnification, pitting and preferential inter-splat corrosion of the coating occurred under these conditions. As can be seen from the figure, in the defect area, due to the existence of a gap, the corrosive medium accumulates at the defect, and there is only a small amount of medium and layer components at the layer position (no defect), indicating that the layer can effectively protect the matrix material.

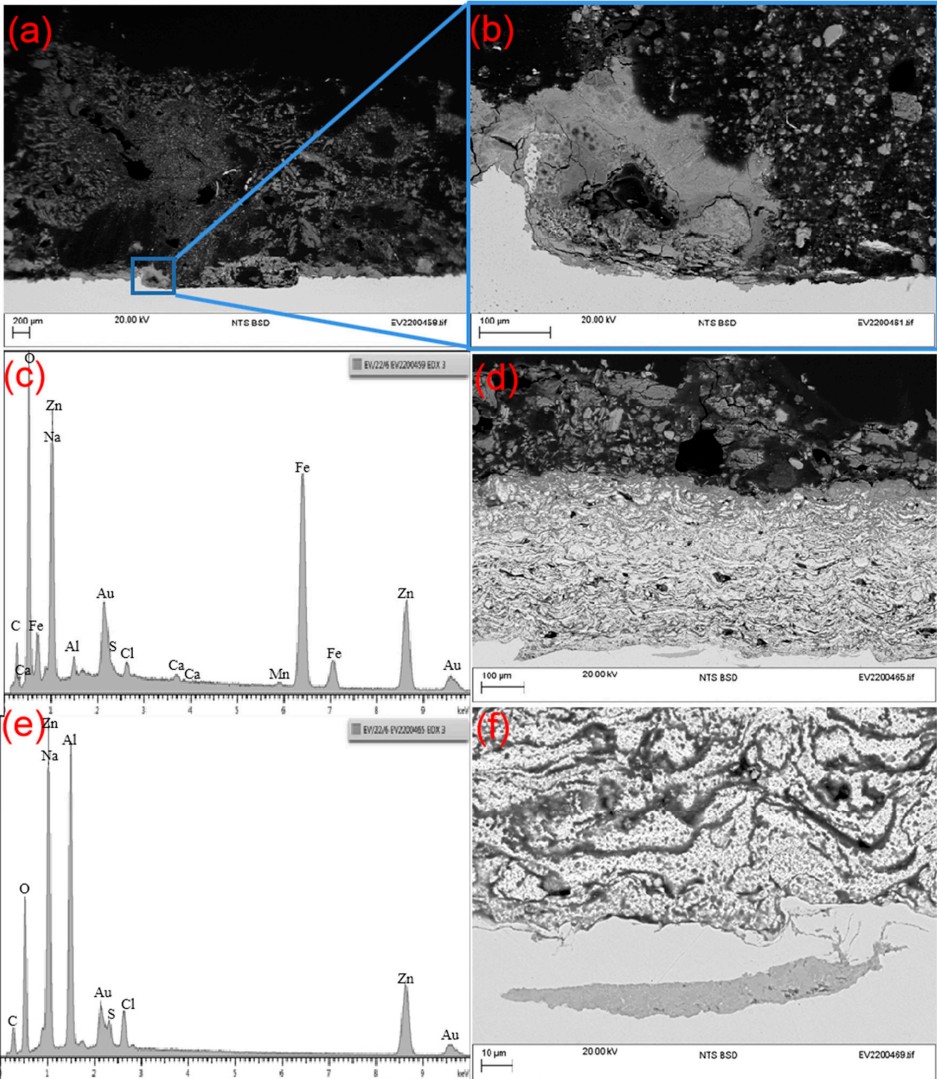

**Figure 7.** SEM micrograph of a Zn-Al coating specimen showing microstructural features after 3 months of testing. (**a**) The coatings near the scribed region; (**b**) enlarged diagram of (**a**); (**c**) the EDS analysis of the corroded area; (**d**) SEM of the faraway scribe region; (**e**) the EDS analysis of the corroded layer; (**f**) higher magnification.

Figure 8a shows the morphology of the Zn-Al coating with paint in the scribed region. As can be seen from the figure, the coating structure has the typical layered structure characteristics, and there are no coarse through pores after sealing. It can be seen from the figure that there is very little coating left in this area. There are copious amounts of corrosion products visible in and around the scribe. Figure 8b shows the EDS analysis near the scribe region. It can be seen from the figure that the area contains Fe, Zn, O, C, and some constituents of the corrosive fluid used in the test. The corrosion products away from the substrate in the scribe seem to be a mixture of Zn, C, O, Al, and some constituents of the corrosive fluid used, as shown in Figure 8c. In regions away from the scribe, the coating seems to have suffered corrosion damage; however, the overall coating remains

intact. Some parts of the coating are beginning to show cracks, as shown in Figure 8d. Under continuous corrosion conditions, discontinuous cracks appear within the coating, as indicated by the black arrow. At the same time, some pit-like morphology appeared on the surface of the coating. Compared with the Zn coating, the corrosion products of the Zn-Al coating are denser in the near-scribed region after the sealing treatment, and the corrosion products of Al can further hinder the corrosion, and there is no obvious corrosion sign far from the defect position. The sealing played an effective protective role against osmosis-induced external corrosion [35]. This is similar to what other researchers have found [24].

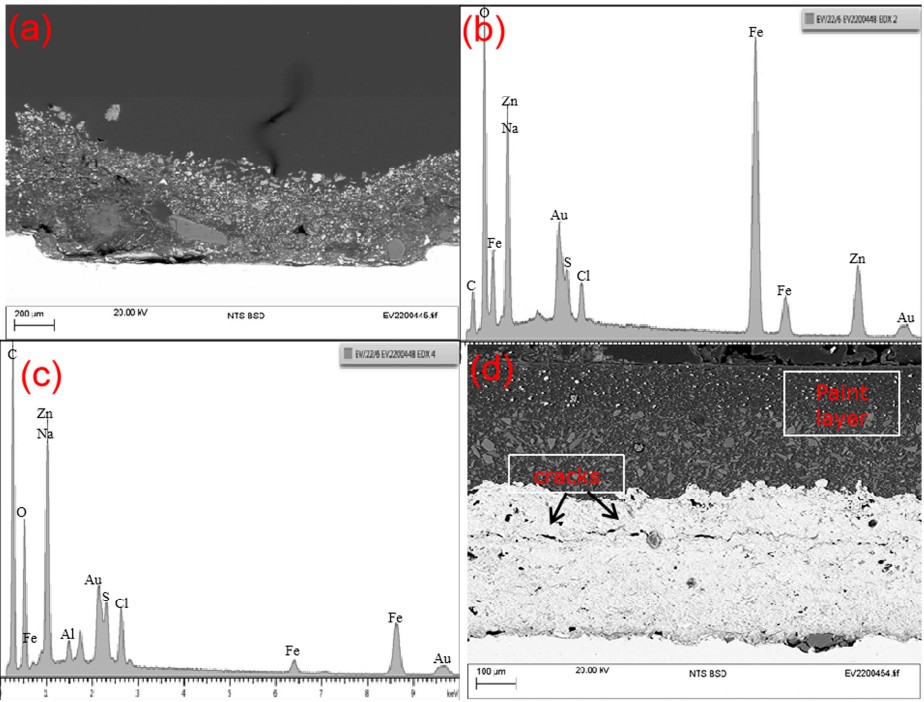

**Figure 8.** SEM micrograph of a Zn-Al coated with paint specimen showing microstructural features after 3 months of testing. (**a**) Zn-Al coating with paint in the scribed region; (**b**) EDS analysis near the scribe region; (**c**) EDS analysis away from the scribe; (**d**) the SEM of the cross section.

### 3.3. Full Immersion and Electrochemical Monitoring

Figure 9 shows the open circuit potential of Zn/Al in the matrix steel and Zn with different spraying thicknesses. The black curve is the matrix steel. It can be seen from the figure that the OCP value of the Zn-Al coating does not seem to depend on the thickness of the coating; the OCP value is similar and does not change with the thickness of the coating, indicating that the Zn-Al coating is relatively stable. These coatings seem to show OCP values between 920 and 940 mV (Ag/AgCl) after 3 months of testing. However, the OCP values of Zn seem to depend on the coating thickness. In the first week of testing, the OCP values of the thick and thin Zn coatings were very similar, but after the first week, the OCP of ~0.15 mm Zn coatings started to become less negative. The likely explanation is the excessive corrosion of Zn in the electrolyte, which resulted in the loss of effective protection of steel by the Zn coating. After 2 weeks of testing, the OCP values were approximately 720 to 740 mV (Ag/AgCl). This is not too different from the OCP values observed for bare steel in the test solution. After 3 weeks of testing, the OCP of thin (~0.15 mm) Zn coatings started to become even less negative and stabilized around 660 to 680 mV (Ag/AgCl) after 3 months of testing. The thicker (~0.3 mm) Zn coatings remained at potentials around 1060 mV after 3 months of exposure (Figure 9b). The main reason for this phenomenon is that, compared with Zn coating, Zn/Al coating shows a better protective effect, and the good performance of Zn/Al coating has no significant correlation with thickness. The

voltage fluctuation after three weeks is very small, and the change process can be further discussed in other ways, such as corrosion rate in Section 3.4.

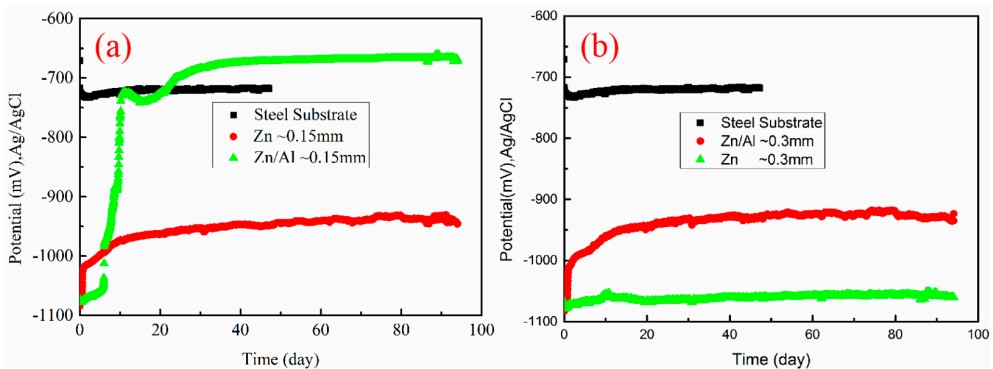

**Figure 9.** Open circuit potential of coatings: (**a**) 0.15 mm and (**b**) ~0.3 mm.

### 3.4. Corrosion Rate

In the macroscopic corrosion observation in Section 3.1, the matrix material has serious corrosion compared with Zn and Zn/Al coatings, so the corrosion rate of the matrix material is not discussed in this part. In a polarization resistance experiment, the current versus applied voltage is recorded as the voltage is swept over a small range of potential. The current is converted to current density knowing the exposed area of the specimen. A numerical fit of the curve (as shown in Figure 10) at zero polarization yields a value for the polarization resistance ($R_p$).

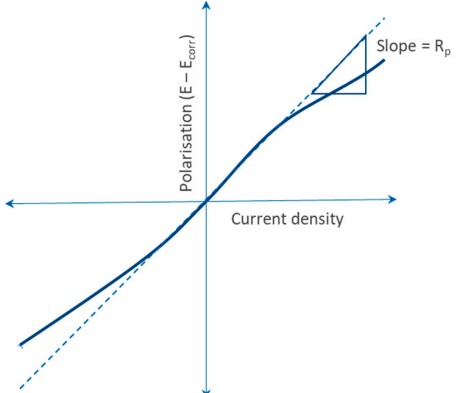

**Figure 10.** The linear polarization plot.

However, polarization resistance data does not give corrosion rate data. To obtain corrosion rate data, Tafel parameters are required, as explained in Equations (1) and (2) in Section 2.3. To obtain Tafel parameters, the specimen is polarized to ±250 mV from OCP (starting from the cathodic side and then going to the anodic side), and the current density is recorded. The final result is shown in Figure 11. The slopes of the anodic and cathodic branches of the potentiodynamic polarization scan give the values of the Tafel constants. The Tafel parameters obtained for the coatings tested in the program are given in Table 3. Zn and Zn/Al have almost the same $b_c$ value; however, Zn/Al ($b_a$) is larger than Zn ($b_a$).

**Table 3.** Tafel parameters for the coatings tested.

| Alloy | $b_a$ | $b_c$ |
|:-----:|:-----:|:-----:|
| Zn | 31.47 | 29.93 |
| Zn-Al | 39.52 | 29.45 |

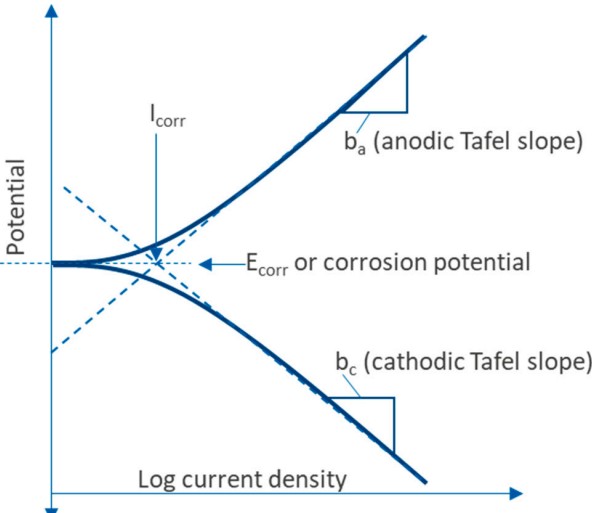

**Figure 11.** A schematic showing the anodic and cathodic polarization curves.

Simultaneously, the corrosion rate of the coatings was calculated and presented in Figure 12. Most of the coatings do not show significant variation in corrosion rate with changes in coating thickness, except for Zn coatings. Thin Zn coatings (~0.15 mm) show corrosion rates of 2–5 mm/y in the first two weeks of testing (Figure 12a); this is similar to the OPC experiment results. However, similar behavior was seen for the thicker Zn coatings (Figure 12b). After the initial settling-in period, the thicker Zn coatings showed a corrosion rate of up to 1 mm/y in the first week of testing, and then the corrosion rates reduced to values below 0.2 mm/y. Significant scatter in the Zn-Al corrosion rate data is also evident (Figure 12b). Figure 10c,d shows the ordinate (corrosion rate) below 0.1 mm/y. The scatter in the Zn corrosion data is also clear. The corrosion rates for Zn were between 0.02–0.06 mm/y for thin coatings and 0.04–0.08 mm/y for thick coatings after 3 months. The corrosion rate of thin Zn-Al coatings was in the range of 0.01 mm/y to 0.02 mm/y after 3 months of testing (Figure 12c). The values show significant scatter for thicker coatings, but the corrosion rates are tending to the values obtained for thinner coatings. In general, Zn-Al coatings are more stable and have better corrosion resistance.

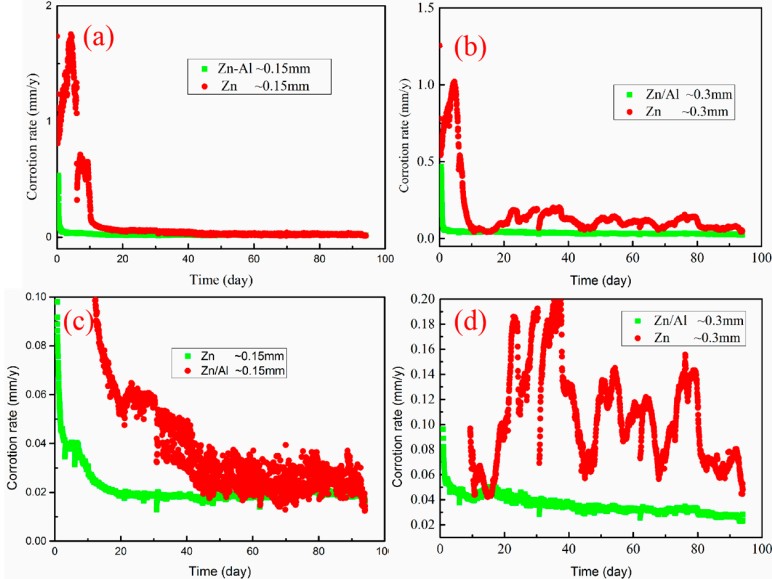

**Figure 12.** Corrosion rate of coatings (**a**) 0.15 mm, (**b**) 0.3 mm, (**c**) 0.15 mm with different scales, and (**d**) 0.3 mm with different scales.

### 3.5. Adhesion Test

According to two different grit types employed for the surface preparation: chilled iron and alumina, the adhesion strength of the coating was tested. The test results are shown in Figure 13. Figure 13a,b shows the comparison of bonding strengths of two grit types (chilled iron and alumina) before and after the salt spray test, and Figure 13c shows the comparison of bonding strengths before and after immersion, where the ordinate is the bonding strength. For better accuracy, five samples (different colors) were selected for the same parameter. As can be seen from the figure, the adhesion strength of the coatings varied depending on the coating composition. The Zn and Zn85-Al15 coatings blasted with brown alumina prior to spraying showed similar adhesion values (~8 MPa). Zn coatings prepared on steel blasted with chilled iron also showed similar average values. However, the Zn85-Al15 coatings showed different adhesion values depending on the blasting media used. Zn85-Al15 prepared on steel blasted with brown alumina showed average adhesion values of ~8 MPa (range ~6–9 MPa), while the ones prepared using chilled iron showed adhesion values exceeding 10 MPa (range ~10–11 MPa). As the average was taken from five test dollies, the difference in the adhesion values is clear.

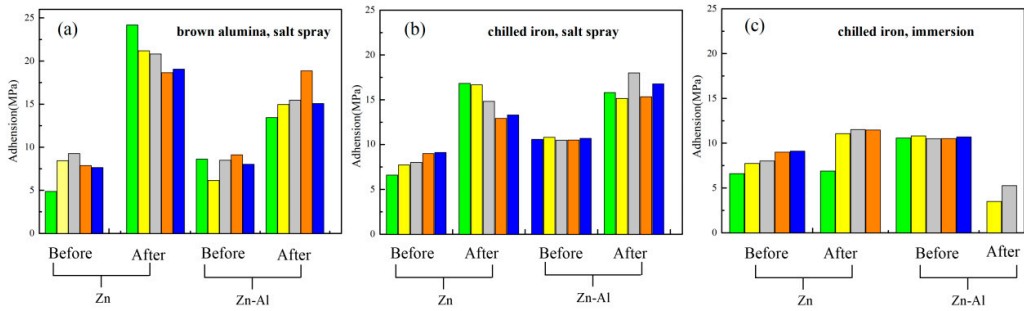

**Figure 13.** Adhesion of thermal spray coatings before and after testing. (**a**) Brown alumina salt spray; (**b**) chilled iron salt spray; (**c**) chilled iron immersion.

The adhesion values of the Zn and Zn85-Al15 coatings seem to have increased after salt spray testing (Figure 13a,b). The Zn coatings prepared on steel blasted with brown alumina and chilled iron showed adhesion values of ~21 MPa and ~15 MPa, respectively (Figure 13a). While the Zn85-Al15 coatings showed adhesion values of ~16 MPa irrespective of the blast media used. (except for one dolly showing lower adhesion, Figure 13b). In the salt spray experiment, the bonding strength increased during corrosion due to different matrix roughnesses [36,37]. The failure mode was cohesive/adhesive in the case of Zn, but cohesive in the case of Zn-Al showed variability in the nature of failure and ranged from predominantly adhesive in some cases to cohesive in others.

An immersion test was carried out on specimens blasted with chilled iron prior to spraying. The Zn specimens showed adhesion values between 6 and 12 MPa with an average of ~10 MPa. Zn85-Al15 coating had the lowest adhesion values (~0–5 MPa), with one test dolly failing before any load was applied for the pull-off test (Figure 13c).

### 3.6. Corrosion Mechanism

The corrosion mechanism of zinc coatings is shown in Figure 14. The corrosion mechanism of Zn coatings can be divided into four stages. Firstly, the oxide film formed by the coating in dry air is destroyed by erosive chloride ions in the corrosive solution. Secondly, the damage caused by strong adsorbable chloride ions on the coating surface causes partial exposure of the passivation film. There is a potential difference between the bare coating and the intact passivation film, and the corrosion solution acts as an electrolyte, speeding up the corrosion process. The formation of the corrosion microcell causes pitting on the surface of the coating, and corrosive substances such as solution ions, oxygen, and water penetrate into the interior of the coating through corrosion pits. Thirdly, as the corrosion progresses, corrosion products accumulate in coating defects and pits,

which can plug the pores and inhibit the corrosion. Zn coating can produce more corrosion products during short-term immersion corrosion, such as ZnO, $Zn_5(OH)_8C_{12} \cdot H_2O$, etc. These corrosion products will plug the pores, resulting in a continuous decline in corrosion current. Fourthly, after a long time of corrosion, the corrosive medium penetrates through the coating to the substrate, and the potential difference between the coating and the substrate promotes the occurrence of electrochemical reactions, while the coating has a lower corrosion potential, so it acts as an anode in corrosion and plays a cathodic protection role on the substrate metal.

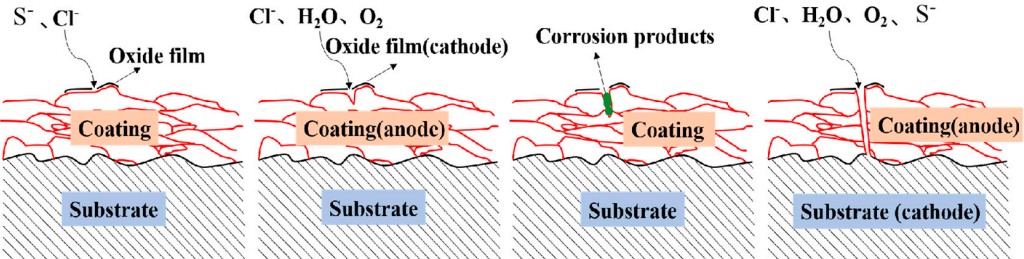

**Figure 14.** Corrosion mechanism of Zn coatings.

The corrosion mechanism of Zn-Al coatings is shown in Figure 15. The corrosion mechanism of the Zn-Al alloy coating is as follows: Firstly, similar to the Zn coating, the coating forms an oxide film on the surface, which is destroyed by corrosive ions. Subsequently, because zinc potential is more negative than aluminum potential, galvanic corrosion forms on the surface. The Zn-rich area with more negative potential is preferentially corroded as the anode. After the Zn dissolution reaction, Zn corrosion products are quickly generated on the coating surface, blocking pores. When Zn corrodes to a certain extent, Al also begins to corrode. Finally, when the corrosive medium reaches the substrate, the coating can also provide cathodic protection for the substrate [38].

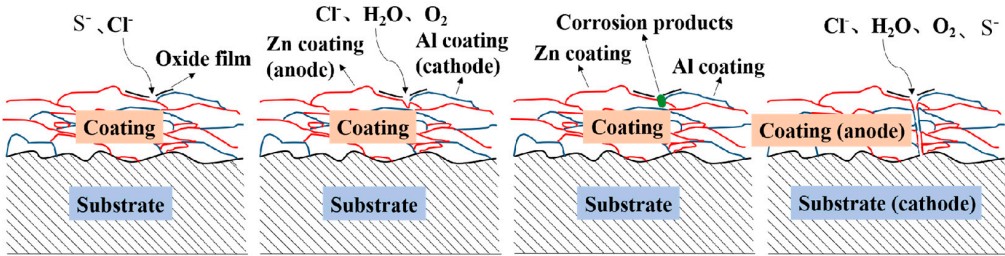

**Figure 15.** Corrosion mechanism of Zn-Al coatings.

## 4. Conclusions

This paper compares the corrosion resistance of Zn and Zn/Al coatings in the project under two different conditions at 40 °C: (1) salt spray in 0.05 wt% NaCl and 0.35 wt% $(NH_4)_2SO_4$ for 3 months under a 1 h spray and 1 h dry cycle, and (2) constant 3-month full immersion in 0.5 wt% NaCl and 3.5 wt% $(NH_4)_2SO_4$. The hole sealing treatment was carried out by Conventional Impregnation Sealing (CIS). The corrosion of the coating in the defect environment is studied, and the performance of the two coatings in various corrosion environments is compared. Based on the systematic experimental studies performed in this study, the following suggestions and conclusions can be drawn:

(1)    After sealing treatment, the defect-free Zn and defect-free Zn-Al coatings have a good application prospect in corrosive environments. After the corrosion of the coating, the corrosion mechanisms of the two coatings are similar in the earlier stage, and the corrosion products of the Zn-Al coating block the corrosion channel in the later stage, resulting in a decrease in the corrosion rate.

(2)    The corrosion resistance of Zn coatings is positively correlated with the coating thickness, while that of Zn/Al coatings is not. After 90 days of immersion, the OCP

of 0.15 mm Zn coating is about −670 mV and that of 0.3 mm Zn coating is about −1050 mV, while the OCP of Zn-Al coating is more stable at about −950 mV.

(3) The corrosion rate of Zn85/Al15 coating at 300 μm is about 0.03 mm/y, and the corrosion rate of Zn coating is higher than Zn85/Al15 coating and is unstable.

(4) The Zn-Al coating exhibits good bonding strength with the substrate in salt spray and immersion medium. The Zn coatings prepared on steel blasted with brown alumina and chilled iron showed adhesion values of ~21 MPa and ~15 MPa, respectively.

(5) In general, the protection effect of the coating on the steel substrate can be significantly enhanced by painting the surface of the Zn-Al coating. It provides good protection, even if there are defects. The long service life of the matrix steel can be realized by spraying the Zn-Al coating.

**Author Contributions:** Conceptualization, B.L. and Z.L.; methodology, J.H., J.B. and H.J.; investigation, Y.T., Z.Z. and S.L. All authors have read and agreed to the published version of the manuscript.

**Funding:** This research was financially supported by the Science and Technology Project of China Southern Power Grid Co., Ltd. (Grant No. GZKJXM20191302).

**Institutional Review Board Statement:** Not applicable.

**Informed Consent Statement:** Not applicable.

**Data Availability Statement:** The data presented in this study are available upon request from the corresponding author.

**Conflicts of Interest:** The authors declare no conflict of interest.

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
