# Peer review of "Effect of Sealing Treatment on Corrosion Resistance of Arc-Sprayed Zn and Zn85-Al15 Coatings"

_coatings, doi:10.3390/coatings13061063_

Round 1
Reviewer 1 Report (Previous Reviewer 1)
I have reviewed this article before. Last time it was rejected by the editor. I should note that this time the authors were very attentive to the comments made earlier by the reviewers. Last time, their attitude to the comments of the reviewers was dismissive. This version of the article has been significantly improved. The quality of drawings and graphs has improved significantly. There was a detailed explanation of the results obtained. The conclusions indicate the specific values obtained in the study. The abstract reflects the essence of the work. The research results can be useful in industrial production.
All the comments I made earlier have been taken into account and I have no comments for this version of the article.
Author Response
I have reviewed this article before. Last time it was rejected by the editor. I should note that this time the authors were very attentive to the comments made earlier by the reviewers. Last time, their attitude to the comments of the reviewers was dismissive. This version of the article has been significantly improved. The quality of drawings and graphs has improved significantly. There was a detailed explanation of the results obtained. The conclusions indicate the specific values obtained in the study. The abstract reflects the essence of the work. The research results can be useful in industrial production.
All the comments I made earlier have been taken into account and I have no comments for this version of the article.
Thanks!
Reviewer 2 Report (New Reviewer)
It is an interesting manuscript, but must undergo minor revision before acceptance in this reputed journal.
1. Please revise the abstract to include more specific information about the results obtained and their implications.
2. The introduction provides a brief background on the topic but lacks a clear statement of the research problem or objective.
3. Please provide a more focused introduction, clearly stating the research objective and its significance.
4. methods section should be expanded to provide more detailed descriptions of the experimental procedures, including the materials used, sample preparation, and testing conditions.
5. Please provide specific information about the morphological analysis techniques, chemical composition analysis methods, and details of the electrochemical studies conducted.
6. Additionally, mention the sample size or number of specimens used in the study.
7. The results section provides a brief summary of the findings but lacks specific data or quantitative information to support the claims made.
8. Include specific information about the need for sealing treatment of arc-sprayed Zn and Zn85-Al15 coatings and the importance of studying their corrosion resistance in a simulated defect environment.
9. Please provide a more comprehensive review by discussing the strengths and limitations of previous studies, identifying research gaps, and explaining how this study contributes to filling those gaps.
10. Explain the purpose of the macroscopic and microscopic morphology analyses and how they relate to the study. Use clearer and more concise language to describe the results. Avoid overly complex sentences and use bullet points or numbered lists to present key observations.
11. Provide more explanation or interpretation of the results. For example, discuss the significance of the corrosion products observed and their implications for the performance of the coatings.
12. Consider adding references to previous studies or relevant literature to support your findings and provide context for the discussion.
13. The research gap is not explicitly stated.
14. Please clearly state the research gap that this study aims to address, such as the lack of comprehensive studies considering multiple parameters (coating thickness, porosity, sealing treatment) and the need to simulate a defect environment.
15. Provide clear descriptions of the observed corrosion morphology, the composition of corrosion products, and the electrochemical measurements obtained.
16. The discussion section should provide a more in-depth analysis and interpretation of the results, comparing them with relevant literature.
17. Analyze the significance of the observed differences in corrosion resistance between the sealed and unsealed coatings.
18. The conclusion should summarize the main findings of the study and their implications concisely.
19. Highlight the practical implications and potential applications of the research.
20. Some recent literature should be added such as Current Nanoscience 18, no. 2 (2022): 203-216 https://doi.org/10.2174/1573413717666210216120741, Applied Sciences 13, no. 2 (2023): 730 https://doi.org/10.3390/app13020730
Minor editing of English language required
Author Response
Comments and Suggestions for Authors
It is an interesting manuscript, but must undergo minor revision before acceptance in this reputed journal.
- Please revise the abstract to include more specific information about the results obtained and their implications.
A new modification was made to the abstract to supplement the conclusions and increase the significance. The anti-corrosive properties of the coatings with or without sealing were compared. The composite coating and the sealing treatment have dual protection mechanism for the matrix. The corresponding modifications to the abstract are highlighted.
- The introduction provides a brief background on the topic but lacks a clear statement of the research problem or objective.
Although many studies have been carried out on zinc-aluminum coatings, few people have analyzed coatings with defects and compared coating properties under multiple parameter conditions, such as coating thickness, acidic medium, sealing treatment etc.The clear statement of the research problem or objective are added in Line104-Line113. In this study, we focus on the combination of corrosive medium and defect environment to analyze the corrosion behavior of different coatings in defect environment. The corrosion mechanism was further analyzed.
- Please provide a more focused introduction, clearly stating the research objective and its significance.
Many researchers have analyzed the corrosion conditions in the corrosive environment, and the effect of sealing treatment on corrosion resistance of Zn-Al alloy coating. Few researchers analyze the defects combined with the corrosive environment. It is not comprehensive to consider only the corrosive medium without considering the defect problem. There is a certain difference with the actual application situation. In practice, corrosive media and defects (gaps, holes) are mutually present. And the analysis of corrosion mechanism of single layer and composite coating is less. Therefore, it is of great value to systematically study the hole sealing performance of arc spray Zn and Zn/Al coating for long-term corrosion protection and to simulate the defect environment and analysis of corrosion mechanism of coating. The above content is added in the introduction.
- methods section should be expanded to provide more detailed descriptions of the experimental procedures, including the materials used, sample preparation, and testing conditions.
The defect simulation method, the material used for coating, the experimental temperature and other parameters are added to the method.For the coating materials used are pure zinc and pure aluminum, the simulation of defects is artificial notching. Other relevant experimental conditions and steps are in 2.2. The 2.2 includes sample preparation and experimental parameters
- Please provide specific information about the morphological analysis techniques, chemical composition analysis methods, and details of the electrochemical studies conducted.
The coating defects, surface morphology of the as-sprayed coatings and coatings after salt spray were examined using a field emission scanning electron microscope (FE-SEM) operated at 15kV (JEOL JSM-7600F). This instrument was equipped with energy-dispersive x-ray spectroscopy (EDS) tool for elemental analysis. The error of SEM and EDS is small, which has little influence on the final analysis result. Potentiodynamic polarization tests were performed after 1500 s open-circuit monitoring. An overpotential of -0.2 to 0.2 V (v.s. OCP) was appliedwith a scanning rate of 0.3 mV/s. To ensure reproducibility all tests were performed at least three times. Details of the above test methods are added in 2.1 and 2.4
- Additionally, mention the sample size or number of specimens used in the study.
The mention the sample size or number of specimens are added in methods.In order to ensure the reliability of the data, three sets of parallel tests were used for analysis.
- The results section provides a brief summary of the findings but lacks specific data or quantitative information to support the claims made.
In the section of corrosion rate, the specific corrosion rate of two coatings in different time periods is added. The macroscopic and microscopic morphology after corrosion can intuitively judge the degree of corrosion, and the quantitative analysis of corrosion mainly depends on the corrosion rate and corrosion potential. The corrosion rates for Zn were between 0.02-0.06mm/y for thin coatings and 0.04-0.08mm/y for thick coatings after 3 months. The corrosion rate of thin Zn-Al coatings were in the range 0.01mm/y to 0.02mm/y after 3 months of testing. The binding strength under different parameters was supplemented. However, the corrosion products are more inclined to the morphology analysis, and the analysis of corrosion products is completed in the micro zone.
- Include specific information about the need for sealing treatment of arc-sprayed Zn and Zn85-Al15 coatings and the importance of studying their corrosion resistance in a simulated defect environment.
After the sealing treatment, the matrix material can realize a double protection mechanism, can significantly reduce the porosity of the coating, and the sealing treatment material can also delay the corrosion.In the actual application process of materials, defects (holes, cracks, crevices, etc.) are common, and the corrosion situation under the simulated defect environment can be more realistic.
- Please provide a more comprehensive review by discussing the strengths and limitations of previous studies, identifying research gaps, and explaining how this study contributes to filling those gaps.
Most previous studies did not consider the influence of defects on corrosion, and there were few studies on the corrosion mechanism of single metal coating and composite coating. In this experiment, corrosion products and corrosion conditions at defects were studied by simulating cracks and other defects. For example, L104-L113
- Explain the purpose of the macroscopic and microscopic morphology analyses and how they relate to the study. Use clearer and more concise language to describe the results. Avoid overly complex sentences and use bullet points or numbered lists to present key observations.
The macroscopic morphology analysis can directly describe the product aggregation and pitting distribution after corrosion. Direct contrast between sealed and unsealed treatment. The macroscopic morphology analysis can directly describe the morphology of the coating after corrosion, and the corrosion situation can be roughly understood through the color change and the accumulation of products.The distribution of corrosion products at the defect location and the defects in the pores of the longitudinal section of the coating can be analyzed by the microscopic morphology analysis. Microstructure analysis can further analyze the morphology of corrosion products. 3.1 The original complex numbering simplification, make the language more simple.
- Provide more explanation or interpretation of the results. For example, discuss the significance of the corrosion products observed and their implications for the performance of the coatings.
In the results part, the precipitation of corrosion products is explained, and the corrosion condition of coating with or without sealing treatment and with or without defects is analyzed and added in 3.1 and 3.2.
- Consider adding references to previous studies or relevant literature to support your findings and provide context for the discussion.
Reference 24.25.31.34 is added. They are the comparison of properties of Zn-Al coatings before and after hole sealing treatment, and the effect of defects on corrosion
- The research gap is not explicitly stated.
Compared with other coating corrosion prevention, this study focuses on the combination of corrosive medium and defect environment, and analyzes the corrosion mechanism of single layer and composite coating.
- Please clearly state the research gap that this study aims to address, such as the lack of comprehensive studies considering multiple parameters (coating thickness, porosity, sealing treatment) and the need to simulate a defect environment.
The purpose of the research and the questions to be explained are added to the introduction. Multiple parameters of the coating were analyzed comprehensively, and a more realistic defect environment was added. Finally, the corrosion mechanism of the coating was analyzed.
- Provide clear descriptions of the observed corrosion morphology, the composition of corrosion products, and the electrochemical measurements obtained.
The morphology and electrochemistry are supplemented, and the macroscopic morphology and microstructure analysis, corrosion products and electrochemical measurements are further described. The corresponding additions are in 3.1, 3.2, 3.3, 3.4
- The discussion section should provide a more in-depth analysis and interpretation of the results, comparing them with relevant literature.
The dual protection mechanism of sealing and coating was summarized, and the corrosion products were analyzed. The corrosion mechanism of other composite coatings is compared. Meanwhile, the corrosion mechanism is compared, Reference 38 is cited for comparative corrosion mechanism.
- Analyze the significance of the observed differences in corrosion resistance between the sealed and unsealed coatings.
For the unsealed coating, the protection mechanism is realized by the coating, which first corrodes the coating to protect the matrix material. In the case of sealing treatment coating, direct contact between corrosive medium and coating and substrate can be avoided and multiple protection mechanisms can be realized
- The conclusion should summarize the main findings of the study and their implications concisely.
(1), (4) and (5) in the conclusion are rewritten to make them more concise and highlight the significance of this experiment
- Highlight the practical implications and potential applications of the research.
The experiment simulated acid rain environment by configuring corrosion solution with different concentrations, and demonstrated the corrosion performance and mechanism of the two coatings by simulating defects. Through the experiment, the spraying protection of matrix steel is realized and its long life application is realized, which has a good reference value for the protection of matrix steel.
- Some recent literature should be added such as Current Nanoscience 18, no. 2 (2022): 203-216 https://doi.org/10.2174/1573413717666210216120741, Applied Sciences 13, no. 2 (2023): 730 https://doi.org/10.3390/app13020730
These two literatures are cited in the article, reference 22.23.
Reviewer 3 Report (New Reviewer)
The manuscript studies the effect of sealing treatment on the corrosion resistance of arc-sprayed Zn and Zn85-Al15 coatings. The manuscript has serious flaws and it cannot be considered for publication. The following comments should be considered:
1- The language of the manuscript is not proper for publication and should be revised by a native English speaker.
2- The introduction does not show the motivation of the work. There are several studies on similar topics and the authors should indicate why this study is necessary.
3- The authors should in detail review the effect of defects on the corrosion behaviour of the Zn coating and their relationship with the corrosion behaviour. Then, they can discuss and complete and discuss the results. Please cite the following manuscript and explain the effect of defects.
https://doi.org/10.1016/j.elecom.2021.107169
4- The reason for the instability of corrosion rate results in Zn coating should be explained in comparison with Zn-Al.
5- In figures 14 and 15, the authors describe several elements affecting the corrosion mechanisms, however, there is not enough evidence for this purpose. Therefore, please remove them or explain in a logical way.
The quality of the language of the manuscript is poor and should be revised by a native English speaker.
Author Response
Comments and Suggestions for Authors
The manuscript studies the effect of sealing treatment on the corrosion resistance of arc-sprayed Zn and Zn85-Al15 coatings. The manuscript has serious flaws and it cannot be considered for publication. The following comments should be considered:
1- The language of the manuscript is not proper for publication and should be revised by a native English speaker.
New changes were made to the language of the manuscript to meet the publication requirements of the journal.
2- The introduction does not show the motivation of the work. There are several studies on similar topics and the authors should indicate why this study is necessary.
Many researchers have analyzed the corrosion conditions in the corrosive environment, and the effect of sealing treatment on corrosion resistance of Zn-Al alloy coating. Few researchers analyze the defects combined with the corrosive environment. It is not comprehensive to consider only the corrosive medium without considering the defect problem. There is a certain difference with the actual application situation. In practice, corrosive media and defects (gaps, holes) are mutually present. Therefore, it is of great value to systematically study the hole sealing performance of arc spray Zn and Zn/Al coating for long-term corrosion protection and to simulate the defect environment and analysis of corrosion mechanism of coating. The above content is added in the introduction.
3- The authors should in detail review the effect of defects on the corrosion behaviour of the Zn coating and their relationship with the corrosion behaviour. Then, they can discuss and complete and discuss the results. Please cite the following manuscript and explain the effect of defects.
Related literature has been added on the effect of defects on coating properties. Defects will accelerate the corrosion process, but corrosion under defects is more practical.The literature study is cited in reference 25
4- The reason for the instability of corrosion rate results in Zn coating should be explained in comparison with Zn-Al.
The reason for the corrosion instability of zinc coating is that the generated corrosion product is non-dense and unstable, which leads to the corrosion solution being able to further penetrate the coating into the matrix, resulting in further corrosion of the matrix. On the other hand, the corrosion products of zinc-aluminum coating are denser, and the fine corrosion products can further plug the formed holes and further hinder corrosion
5- In figures 14 and 15, the authors describe several elements affecting the corrosion mechanisms, however, there is not enough evidence for this purpose. Therefore, please remove them or explain in a logical way.
In order to more intuitively illustrate the differences between single coating and composite coating, FIG. 14.15 is used to describe the corrosion differences between the two coatings. The dense products after corrosion of zinc-aluminum coating can further hinder the corrosion channel and delay the corrosion.It can be used as reference for corrosion mechanism in corrosive medium environment.
Round 2
Reviewer 3 Report (New Reviewer)
The answers to the comments and the corresponding revisions are satisfactory. I recommend publication in the current form.
The language of the manuscript should be double-checked before publication.
This manuscript is a resubmission of an earlier submission. The following is a list of the peer review reports and author responses from that submission.
Round 1
Reviewer 1 Report
The paper is devoted to the study of Zn and Zn/Al coatings obtained by electric arc deposition. The research topic is relevant and promising. I have previously reviewed this document. The authors significantly improved the presentation of the results.
However, I have some comments on this paper:
1. In Figures 7 and 10, markers are poorly chosen. The curves merge and are indistinguishable. It is not possible to analyze the presented results. What is the difference between curves "1" and "2" in figures 7 and 10?
2. What do the colors in the bar charts in Figure 9 mean? There are no explanations on the figures. Explain to readers of the article.
Author Response
Comments and Suggestions for Authors
The paper is devoted to the study of Zn and Zn/Al coatings obtained by electric arc deposition. The research topic is relevant and promising. I have previously reviewed this document. The authors significantly improved the presentation of the results.
However, I have some comments on this paper:
- In Figures 7 and 10, markers are poorly chosen. The curves merge and are indistinguishable. It is not possible to analyze the presented results. What is the difference between curves "1" and "2" in figures 7 and 10?
Figure 7 and Figure 10 were revised and the data were processed, and the original two curves were changed into one.
- What do the colors in the bar charts in Figure 9 mean? There are no explanations on the figures. Explain to readers of the article.
In order to ensure the reliability of the experiment, five samples (i.e., five colors) were selected for the parameters in the same state, which were respectively the strength of bonding force before and after the salt spray test and immersion.
Reviewer 2 Report
Dear author(s), the manuscript titled ‘Effect of sealing treatment on corrosion resistance of arc sprayed Zn and Zn85-Al15 coatings’, Manuscript ID: coatings-2329469, has some weaknesses that must be significantly improved before any further actions in the processing, if allowed by the Editor.
Please refer to the comments below:
1. Commenting on the ‘Abstract’ section, I would omit to introduce ‘rough’ results but, respectively, only a main, general conclusion at the end of this sentence. The rough results must be provided in the ‘Results and Discussion’ section only and, respectively if required in the ‘Conclusion’ section.
2. According to the previous issue, one general conclusion indicating the novelty proposed should be addressed with more detail. In the current form, the ‘Abstract’ section is too heavy and the main result, in the end, is too poor. From that point of view, the Abstract is not clear.
3. The ‘Introduction’ section, generally, is too heavy that was not separated into different research. In the current form looks like the review was prepared for only a short part of the area of study but, respectively, should be presented more comprehensively.
4. Further to this section, there is a lack of critical review of the current state of knowledge in the ‘Introduction’. Please try to emphasize the meaning of the studies performed in the whole ‘Introduction’ section. Currently, its proposals do not derive from the literature review and are not indicating the lack in the current state of knowledge.
5. In section 2.1 the figure presenting the workpiece should be added by Author(s).
6. In section 2.2 the figure for the workstation is also lacking. Only about the samples after salt spray chambers (like Figure 2 in section 3.1).
7. Equations (1) and (2) should be referenced to the primary sources if received from selected papers or books. In the current form, it is difficult to convey if equations are newly proposed or not or, respectively, if they can be classified as a part of the novelty.
8. Section 2.4 was not properly detailed about the post-test data processing. Firstly, there is no word on the SEM measurement. Secondly, X-ray (EDX) characterisation was not accurately presented as well. The accuracy of both methods was not even mentioned. What about the SEM measurement uncertainty or noise? Please try to emphasize those issues, like referenced in:
(1) https://doi.org/10.1088/2051-672X/3/3/035004
(2) https://doi.org/10.3390/ma14175096
(3) https://doi.org/10.24425/mms.2021.137706
9. The size of Figures 3, 4, 5, 6, 7 and 10 must be improved. It is difficult to receive the number data. Moreover, the font size of the text in all of the Figures should be unified, e.g. ‘(a)’ ‘(b)’… in some Figures (3) is much smaller than others (like in Figure 6).
10. In the ‘Conclusion’ section, one, general idea should be added at the end (last conclusion) indicating the main purpose of the manuscript.
11. All of the items cited in the ‘References’ should be formatted according to the journal template requirements.
12. The full DOI links should be added to the references, if exist.
Generally, the proposed manuscript is interesting but, respectively, must be improved significantly which includes many practical weaknesses.
Some issues make understanding the paper difficult and the reader confused.
Therefore, the manuscript should be improved in a required manner before any further processing for publication in a quality journal as the Coatings is, if allowed by the Editor.
Author Response
Comments and Suggestions for Authors
Dear author(s), the manuscript titled ‘Effect of sealing treatment on corrosion resistance of arc sprayed Zn and Zn85-Al15 coatings’, Manuscript ID: coatings-2329469, has some weaknesses that must be significantly improved before any further actions in the processing, if allowed by the Editor.
Please refer to the comments below:
- Commenting on the ‘Abstract’ section, I would omit to introduce ‘rough’ results but, respectively, only a main, general conclusion at the end of this sentence. The rough results must be provided in the ‘Results and Discussion’ section only and, respectively if required in the ‘Conclusion’ section.
The abstract section has been modified.
- According to the previous issue, one general conclusion indicating the novelty proposed should be addressed with more detail.
It was indicated that the Zn85/Al15 coating prepared by CIS has better corrosion resistance of the matrix steel.
- The ‘Introduction’ section, generally, is too heavy that was not separated into different research. In the current form looks like the review was prepared for only a short part of the area of study but, respectively, should be presented more comprehensively.
In the introduction part, the research status of zinc and zinc-aluminum coatings is summarized. Meanwhile, the sealing experiment is analyzed, and the corrosion mechanism, characteristics, macro and micro defects are analyzed.The above review is closely related to the topic of this paper, and pave the way for the hole sealing treatment, defect corrosion and so on
- Further to this section, there is a lack of critical review of the current state of knowledge in the ‘Introduction’. Please try to emphasize the meaning of the studies performed in the whole ‘Introduction’ section. Currently, its proposals do not derive from the literature review and are not indicating the lack in the current state of knowledge.
P100-109 is the research significance of this paper.
- In section 2.1 the figure presenting the workpiece should be added by Author(s).
The schematic diagram of the workpiece has been added in figure 1.
- In section 2.2 the figure for the workstation is also lacking. Only about the samples after salt spray chambers (like Figure 2 in section 3.1).
The figure for the workstation has been added in 2.2.
- Equations (1) and (2) should be referenced to the primary sources if received from selected papers or books. In the current form, it is difficult to convey if equations are newly proposed or not or, respectively, if they can be classified as a part of the novelty.
Equations (1) and (2) are quoted from references 22-24.
- Section 2.4 was not properly detailed about the post-test data processing. Firstly, there is no word on the SEM measurement. Secondly, X-ray (EDX) characterisation was not accurately presented as well. The accuracy of both methods was not even mentioned. What about the SEM measurement uncertainty or noise? Please try to emphasize those issues, like referenced in:
(1) https://doi.org/10.1088/2051-672X/3/3/035004
(2) https://doi.org/10.3390/ma14175096
(3) https://doi.org/10.24425/mms.2021.137706
In 2.4, SEM measurement is added, and EDX features are described, and its influence on experimental results is analyzed
- The size of Figures 3, 4, 5, 6, 7 and 10 must be improved. It is difficult to receive the number data. Moreover, the font size of the text in all of the Figures should be unified, e.g. ‘(a)’ ‘(b)’… in some Figures (3) is much smaller than others (like in Figure 6).
The size of Figures 3, 4, 5, 6, 7 and 10 have been improved, the font size of the text in all of the Figures are unified
- In the ‘Conclusion’ section, one, general idea should be added at the end (last conclusion) indicating the main purpose of the manuscript.
- All of the items cited in the ‘References’ should be formatted according to the journal template requirements.
The format of the references has been modified according to the journal requirements
- The full DOI links should be added to the references, if exist.
The full DOI links have been added to the references.
Reviewer 3 Report
The effect of sealing treatment on the corrosion resistance of arc-sprayed Zn and Zn85-Al15 coatings is presented in this paper. The comments provided below must be addressed before this paper can be accepted.
1. At the moment only 20 references are added. Although authors have added several references from 2022, it would be better if some references from 2023 could be added.
2. A synthesis between the literature review and current work is lacking. The authors summarize all the references but without creating any bridge between their study.
3. The authors must also clearly state the novelty of this study. It is important to highlight the novel aspects.
4. The details and figure showing the spraying mechanism are missing.
5. The author do not specify any measurement errors? Does the study present perfect results?
6. A detailed schematic of sample preparation and tests conducted should be added in 2.1.
7. Sections 3.1-3.4 lack quantitative details most of the information is qualitative which is difficult to follow.
8. Figures 7 and 10 should be improved.
Author Response
Comments and Suggestions for Authors
The effect of sealing treatment on the corrosion resistance of arc-sprayed Zn and Zn85-Al15 coatings is presented in this paper. The comments provided below must be addressed before this paper can be accepted.
- At the moment only 20 references are added. Although authors have added several references from 2022, it would be better if some references from 2023 could be added.
New references have been added and there are now 32 references, including 2023
- A synthesis between the literature review and current work is lacking. The authors summarize all the references but without creating any bridge between their study.
In p101-105, the relationship between the literature review and the current experiment was added, which integrated the experimental conditions of different spray coating components, coating thickness, hole sealing treatment, defect environmental corrosion and so on.
- The authors must also clearly state the novelty of this study. It is important to highlight the novel aspects.
The current work systematically analyzes the coating of different thickness, and simulates the defect corrosion environment, which is more close to the actual corrosion situation
- The details and figure showing the spraying mechanism are missing.
A picture of the spraying machine has been added as shown in Figure 1.
- The author do not specify any measurement errors? Does the study present perfect results?
There are some errors in the testing process, but the errors are within the controllable range, and the impact on the final experimental results is small, which is acceptable. Note the margin of error in the late process
- A detailed schematic of sample preparation and tests conducted should be added in 2.1.
The detailed schematic of sample preparation and tests conducted was added in 2.1
- Sections 3.1-3.4 lack quantitative details most of the information is qualitative which is difficult to follow.
3.1,3.2,3.3 are the schematic diagram of coating microstructure and EDS spectrum, Is a schematic diagram of the morphology of the coating after corrosion. 3.4 are OCP curves of coatings of different thicknesses. The above experiments can directly represent the change of its performance.
- Figures 7 and 10 should be improved.
The Fig.7 and Fig.10 have been improved.

Round 2
Reviewer 2 Report
Dear Author(s), the final review of the manuscript titled: ‘Effect of sealing treatment on corrosion resistance of arc sprayed Zn and Zn85-Al15 coatings’, Manuscript ID: coatings-2329469.
Comment 1: Commenting on the ‘Abstract’ section, I would omit to introduce ‘rough’ results but, respectively, only a main, general conclusion at the end of this sentence. The rough results must be provided in the ‘Results and Discussion’ section only and, respectively if required in the ‘Conclusion’ section.
Author(s) response: The abstract section has been modified.
Final comment: Please present any modifications.
Comment 2: According to the previous issue, one general conclusion indicating the novelty proposed should be addressed with more detail. In the current form, the ‘Abstract’ section is too heavy and the main result, in the end, is too poor. From that point of view, the Abstract is not clear.
Author(s) response: It was indicated that the Zn85/Al15 coating prepared by CIS has better corrosion resistance of the matrix steel.
Final comment: Any word in the ‘Abstract section was not improved It is weak and respectively, cannot be accepted!
Comment 3: The ‘Introduction’ section, generally, is too heavy that was not separated into different research. In the current form looks like the review was prepared for only a short part of the area of study but, respectively, should be presented more comprehensively.
Author(s) response: In the introduction part, the research status of zinc and zinc-aluminum coatings is summarized. Meanwhile, the sealing experiment is analyzed, and the corrosion mechanism, characteristics, macro and micro defects are analyzed.The above review is closely related to the topic of this paper, and pave the way for the hole sealing treatment, defect corrosion and so on
Final comment: This response and manuscript improvements can be initially accepted.
Comment 4: Further to this section, there is a lack of critical review of the current state of knowledge in the ‘Introduction’. Please try to emphasize the meaning of the studies performed in the whole ‘Introduction’ section. Currently, its proposals do not derive from the literature review and are not indicating the lack in the current state of knowledge.
Author(s) response: P100-109 is the research significance of this paper.
Final comment: There is no required critical review. Lines 100-109 did not increase the motivation of the manuscript.
Comment 5: In section 2.1 the figure presenting the workpiece should be added by Author(s).
Author(s) response: The schematic diagram of the workpiece has been added in figure 1.
Final comment: Even though the figure was improved all of the doubts were not significantly removed. However, the response can be accepted.
Comment 6: In section 2.2 the figure for the workstation is also lacking. Only about the samples after salt spray chambers (like Figure 2 in section 3.1).
Author(s) response: The figure for the workstation has been added in 2.2.
Final comment: Response is accepted.
Comment 7: Equations (1) and (2) should be referenced to the primary sources if received from selected papers or books. In the current form, it is difficult to convey if equations are newly proposed or not or, respectively, if they can be classified as a part of the novelty.
Author(s) response: Equations (1) and (2) are quoted from references 22-24.
Final comment: Response is accepted.
Comment 8: Section 2.4 was not properly detailed about the post-test data processing. Firstly, there is no word on the SEM measurement. Secondly, X-ray (EDX) characterisation was not accurately presented as well. The accuracy of both methods was not even mentioned. What about the SEM measurement uncertainty or noise? Please try to emphasize those issues, like referenced in:
(1) https://doi.org/10.1088/2051-672X/3/3/035004
(2) https://doi.org/10.3390/ma14175096
(3) https://doi.org/10.24425/mms.2021.137706
Author(s) response: In 2.4, SEM measurement is added, and EDX features are described, and its influence on experimental results is analyzed
Final comment: The response ‘The error of SEM and EDX is small, which has little influence on the final analysis result.’ is not acceptable. This issue was not treated seriously but has a significant influence on the measurement results.
Comment 9: The size of Figures 3, 4, 5, 6, 7 and 10 must be improved. It is difficult to receive the number data. Moreover, the font size of the text in all of the Figures should be unified, e.g. ‘(a)’ ‘(b)’… in some Figures (3) is much smaller than others (like in Figure 6).
Author(s) response: The size of Figures 3, 4, 5, 6, 7 and 10 have been improved, the font size of the text in all of the Figures are unified
Final comment: Only formatting issues were fulfilled.
Comment 10: In the ‘Conclusion’ section, one, general idea should be added at the end (last conclusion) indicating the main purpose of the manuscript.
Author(s) response:
Final comment: I cannot catch why the Author(s) did not improve this issue. The ‘Conclusion’ section is one of the most crucial parts of the manuscript. If not appropriately referring to the results presented, the whole manuscript is useless.
Comment 11: All of the items cited in the ‘References’ should be formatted according to the journal template requirements.
Author(s) response: The format of the references has been modified according to the journal requirements
Final comment: They were not modified!
Comment 12: The full DOI links should be added to the references, if exist.
Author(s) response: The full DOI links should be added to the references, if exist.
Final comment: Only this comment was fully responded and modifications were provided.
I feel sorry but the Author(s) did not threaten the comments seriously so, respectively, the revised manuscript was not improved suitably and cannot be accepted.
Please do not waste the time of the reviewer.
Reviewer 3 Report
All required changes have been incorporated.
Author Response
thanks